# Using Motion Cues to Supervise Single-Frame Body Pose and Shape Estimation in Low Data Regimes

**Andrey Davydov**[*]                                                    *andrey.davydov@epfl.ch*
*CVLab, EPFL*

**Alexey Sidnev**                                                        *asidnev@meta.com*
*Meta AI*

**Artsiom Sanakoyeu**                                                    *asanakoy@meta.com*
*Meta AI*

**Yuhua Chen**                                                          *yuhuac@meta.com*
*Meta AI*

**Mathieu Salzmann**                                                    *mathieu.salzmann@epfl.ch*
*CVLab, EPFL*

**Pascal Fua**                                                          *pascal.fua@epfl.ch*
*CVLab, EPFL*

**Reviewed on OpenReview:** *https://openreview.net/forum?id=fUhOb14sQv*

## Abstract

When enough annotated training data is available, supervised deep-learning algorithms excel at estimating human body pose and shape using a single camera. The effects of too little such data being available can be mitigated by using other information sources, such as databases of body shapes, to learn priors. Unfortunately, such sources are not always available either. We show that, in such cases, easy-to-obtain unannotated videos can be used instead to provide the required supervisory signals. Given a trained model using too little annotated data, we compute poses in consecutive frames along with the optical flow between them. We then enforce consistency between the image optical flow and the one that can be inferred from the change in pose from one frame to the next. This provides enough additional supervision to effectively refine the network weights and to perform on par with methods trained using far more annotated data.

## 1 Introduction

Given enough annotated training data, supervised deep-learning algorithms for estimating human body shape and pose using a single camera have become remarkably effective (Moon & Lee, 2020; Choi et al., 2021; Wei et al., 2022). The very recent transformer-based architecture of (Goel et al., 2023) embodies the current state-of-the-art. It is pre-trained on 300 million images and fine-tuned on most SMPL data sets in existence using a GAN discriminator. This makes it an excellent foundation model but, unfortunately, such a large amount of data is not always forthcoming and requires massive amounts of computing power. Furthermore, even though the datasets are publicly available for research purposes, networks trained on them may not be usable commercially for licensing or privacy reasons. Specific licenses often restrict the use of data for commercial purpose, which drives companies to either spend resources for their own data acquisition or to

---

[*]This work was supported in part by the Swiss National Science Foundation and done in part during an internship at Meta AI.

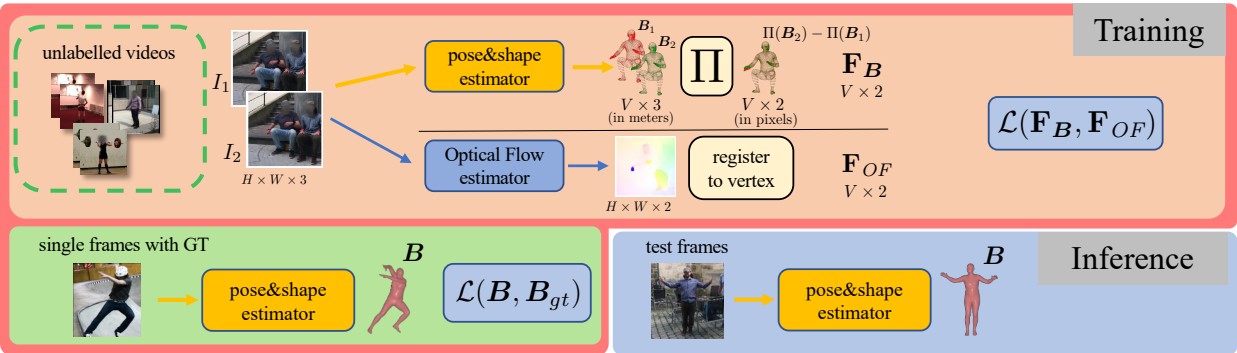

Figure 1: **Method Overview**. Self-supervised training with optical flow guidance (**Top**). Given a pre-trained network that takes single frames as input, we compute SMPL body mesh estimates $B$ in consecutive frames along with the optical flow $\mathbf{F}_{OF}$ between them. This lets us write a loss term based on the consistency of $\mathbf{F}_{OF}$ and of the flow $\mathbf{F}_B$ that can be inferred from the changes in $B$ across the frames. We then minimize a weighted sum of this loss and the one used to pre-train the network (**Bottom left**). At inference time (**Bottom right**), the refined body estimator can run on single images and does not require the optical flow anymore.

develop other approaches that require less supervision. Meanwhile, privacy issues might entail legal risks for further use of data, especially human poses that can be viewed as personal information.

To overcome these issues, there have been many attempts at obtaining good results with less supervision, such as imposing body pose and motion priors (Kanazawa et al., 2018; Davydov et al., 2022; Kocabas et al., 2020; Luo et al., 2020) or creating synthetic training data (Sengupta et al., 2021; Hewitt et al., 2023). Most of these approaches to making the training more data-efficient require external sources of information, such as 2D/3D keypoints or silhouettes, in addition to the images themselves. In addition to the privacy and licensing issues mentioned above, such targeted data acquisition can be labor intensive. This motivates our search for more data-efficient techniques. In this paper, we advocate exploiting a widely available source of information. which has largely been overlooked for training purposes. Since most cameras can acquire video sequences, we propose relying on motion consistency within these sequences to provide additional supervision—without further annotations—and to refine the network weights so that it ultimately delivers better result in *single* images at inference time. In other words, we use the sequences at training time but the refined run-time network does not require them, which is *not* true for current approaches to exploiting motion information (Wan et al., 2021; Li et al., 2022; Wei et al., 2022). This is a design limitation that restricts their generality.

In practice, we start from a potentially poorly trained network—for example because not enough annotated images were available to train it better—that outputs predictions in terms of the SMPL body model (Loper et al., 2015). We then use an off-the-shelf algorithm (Teed & Deng, 2020) to estimate the flow in unannotated video sequences and rely on the SMPL fixed topology to refine the network weights so that its predictions in individual images are consistent with it. Fig. 1 depicts this approach. This does not require changing the architecture of the network we want to refine. Furthermore, the optical flow estimator is trained on synthetic data, such as FlyingThings3D (Mayer et al., 2016) or Sintel (Butler et al., 2012), which has nothing to do with body motion estimation and does not require human annotation. In other words, our approach is flexible, generic, and easy to deploy.

We show that this outperforms the state-of-the-art semi-supervised TexturePose (Pavlakos et al., 2019), a pose and shape estimation technique that exploits color consistency instead of motion. This is significant because TexturePose is the only semi-supervised technique we know of that extracts useful body-shape information from *unlabeled* videos. Furthermore, color and motion-consistency are complementary sources of information. Hence, we show that using them jointly works even better. Even though our main focus is in using only unlabelled videos for additional supervision, we compare against an off-the-shelf 2D keypoint estimator that uses large amount of posing data during its training. Although the keypoints provide a useful signal, our use of optical flow delivers an additional enhancement. Additionally, we demonstrate that

using more unannotated videos during training significantly improves the performance of the model. Finally, even though our approach is designed to operate on single images, with only minor changes, it can exploit sequences at run time and bridge the gap between monocular and video-based pose estimators.

Our method delivers these benefits for various network architectures and our experiments show that our approach further strengthens already strong baseline models relying on different backbones. Thus, our contribution is an approach to using videos to refine a network designed to work on single images, which has never been done before. This lets us start with an insufficiently trained network and refine it without having to provide additional annotated data or to modify it in any way. This makes our method useful in practical settings where not enough annotated data is available to properly train the network in a fully supervised manner.

## 2   Related Work

Our focus is on providing self-supervision for methods that regress SMPL models solely from images, while requiring as little additional information as possible beyond the images themselves. Thus we first examine existing semi-supervised approaches and argue that they require far more additional information than the off-the-shelf optical flow estimator we rely on. We then turn the very few methods that take a similar tack as we do. Finally, we discuss video-based methods, which, while powerful, are not well suited for the single-image setting we address here.

**Semi-supervised Pose and Shape Estimation.**  A common approach to making the training of 3D body pose estimation networks less data intensive is to rely on strategies that lift 2D poses to 3D (Deng et al., 2021; Chen et al., 2019). In the specific case of networks that regress SMPL parameters (Loper et al., 2015), PoseNet3D (Tripathi et al., 2020) uses sequential teacher-student training to do this, while HybriK (Li et al., 2021) relies on an elaborate well-trained 3D pose estimator coupled with complex inverse kinematics computations. Unfortunately, this still requires considerable amounts of 3D and SMPL ground-truth.

In theory, we could also use techniques that are model-agnostic. They rely on camera priors (Wandt et al., 2022), kinematics preservation (Kundu et al., 2020a; Chen et al., 2022), or multi-camera consistency (Rhodin et al., 2018; Gholami et al., 2022). However, they are designed to work under very specific circumstances and lack the generality of what we propose here.

**Optical Flow for Pose and Shape Estimation.**  The central idea of this paper is to rely solely on consecutive images and the optical flow ($OF$) between them to refine a pre-trained network. $OF$ has clearly been used before for human pose estimation, but not in this particular way.

In (Romero et al., 2015), 2D body poses are directly estimated from the video using only optical flow. (Arko et al., 2022) introduces an optimization bootstrapping algorithm to enhance off-the-shelf 3D skeleton body and optical flow estimators at *inference* time. In (Tung et al., 2017), optical flow is used along with keypoints and silhouettes to refine the output of a pretrained SMPL-based body shape estimator. In (Doersch & Zisserman, 2019), a CNN-based body regressor is split in two independently trained segments. The first is trained to estimate $OF$ and 2D heatmaps directly from images using real-world video datasets. The second regresses body parameters explicitly from $OF$ maps and heatmaps, and the supervision is taken from synthetic datasets. The unconditioning of body estimates from the noisy or irrelevant image cues helps close the domain gap between synthetic and real video sequences. However, this approach requires the optical flow at *inference time* and, thus, cannot handle single images. In (Ranjan et al., 2018; 2020), the authors follow up by introducing a novel synthetic dataset containing very clean $OF$ maps paired with image-body data.

**Color Consistency for Pose and Shape Estimation.**  Instead of using $OF$, one can use color consistency. In (Pavlakos et al., 2019), textures of the rendered body avatar are extracted from images, and the networks are trained so that they conform with what is seen in different views. This idea is extended in (Kundu et al., 2020b) by selecting pairs of images with various background colors and taking into account

the visibility of particular body points. Hence, this approach is complementary to ours. We will show that jointly using color consistency and motion information yields better results than using either alone.

**Video-Based Pose and Shape Estimation.** Soon after SMPL body estimators from single images (Kanazawa et al., 2018) appeared, they were extended to handle video inputs and mostly differ in how they encode temporal relations.

A common approach (Kanazawa et al., 2019; Kocabas et al., 2020; Luo et al., 2020; Choi et al., 2021) is to extract per-frame features and then mix them through temporal encoding. HMMR (Kanazawa et al., 2019) exploits 2D poses extracted from videos to predict past and future poses from the current frame. VIBE (Kocabas et al., 2020) uses recurrent layers to mix temporal features and penalizes unrealistic body sequences by adversarial training. In MEVA (Luo et al., 2020), the discriminator is replaced by a VAE-based motion prior, while TCMR (Choi et al., 2021) disentangles static and temporal features to increase temporal consistency. Recent advances in self-attention mechanisms have given rise to a new generation of methods that rely on spatial and temporal transformers, including MAED (Wan et al., 2021), DTS-VIBE (Li et al., 2022) and the state-of-the-art MPS-Net (Wei et al., 2022).

All these methods require large sets of annotated video data, including 2D, 3D and ground-truth SMPL parameters, when available. Furthermore, they are typically designed to take as input a complete video. Hence, they cannot be used for single-frame pose and shape estimation. We demonstrate that, in this setting, our algorithm can exploit temporal context and bridge the gap between monocular and video-based methods.

## 3 Method

In this section, we present our approach to fine-tuning a network by enforcing consistency between the motion flow that can be inferred from detections in consecutive images and the actual optical flow that can be estimated from these images. In Sec. 3.1, we start by describing the standard approach to estimating body pose and shape using a baseline model. In Sec. 3.2, we introduce our proposed *OF*-based technique to refining the baseline given unlabelled videos. Finally in Sec. 3.3, we argue that a minor extension of our single-image based approach allows temporal inference to bridge the gap with video-based methods.

### 3.1 Problem Statement

Let us assume we are given a pre-trained body shape and pose estimation network, which we will refer to as *BL* ("baseline"), from the so-called human mesh recovery family of models (Kanazawa et al., 2018; Kolotouros et al., 2019; Joo et al., 2020). It takes as input a single image $\mathbf{I}$ and outputs a human pose representation in terms of the SMPL (Loper et al., 2015; Bogo et al., 2016) pose and shape parameters, $\boldsymbol{\theta}$ and $\boldsymbol{\beta}$ respectively, along with a camera projection estimate $\boldsymbol{\pi}$. As we are mostly concerned with body shape and pose estimation, camera parameter estimation is of secondary importance. Hence, as in many other approaches, we rely on the weak perspective projection camera model. This follows common practice and provides us with volumetric and semantic body representations we need. We will refer to the estimated body mesh vertices as $\boldsymbol{B}$ of size $V \times 3$.

Our goal is to fine-tune *BL* using the information provided by the optical flow between consecutive video frames, without having to supply any additional information or annotations. Once this is done, *BL* does not need the motion information anymore and can operate on *single* images.

### 3.2 Optical Flow as Weak Supervision

Given two consecutive images $\mathbf{I}_1$ and $\mathbf{I}_2$, we define a loss that quantifies the discrepancy between

- the *OF* map $\mathbf{F}_{OF}^{1\to2}$ from $\mathbf{I}_1$ to $\mathbf{I}_2$ estimated by an off-the-shelf optical flow algorithm (Teed & Deng, 2020),

- the flow map $\mathbf{F}_{\boldsymbol{B}}^{1\to2}$ that can be computed from the vertex displacements between $\boldsymbol{B}_1$ and $\boldsymbol{B}_2$—the SMPL body meshes estimated from both images—projected into $\mathbf{I}_1$ and $\mathbf{I}_2$.

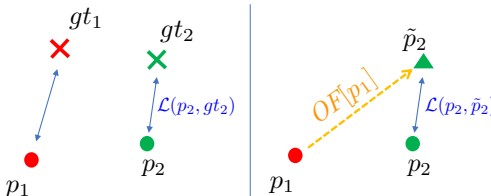

Figure 2: **Point-wise nature of the *OF* motion-alignment loss.** Points $p_1$ and $p_2$ are the predictions for the corresponding frames. If ground-truth labels are given (*left*), then supervision is straightforward. In the absence of ground truth (*right*), we use $p_1$ displaced by the *OF* estimate as a weak label for $p_2$.

Ideally, $\mathbf{F}_{OF}$ and $\mathbf{F}_{B}$ should be equal, and we define

$$L_{OF}^{1\to2} = \|(\mathbf{F}_{B}^{1\to2} - \mathbf{F}_{OF}^{1\to2}[\mathbf{\Pi}_1(\boldsymbol{B}_1)]) \odot M\|_2 \,, \tag{1}$$
$$\mathbf{F}_{B}^{1\to2} = \mathbf{\Pi}_2(\boldsymbol{B}_2) - \mathbf{\Pi}_1(\boldsymbol{B}_1),$$

where $M$ is the intersection of the binary masks of the vertices visible in each frame. These *visibility* masks are $V$-dimensional vectors computed by differentiable rendering of the 3D body mesh. $\mathbf{\Pi}_1$ and $\mathbf{\Pi}_2$ denote the projection of the 3D vertices into the images and $\mathbf{F}_{OF}^{1\to2}[\mathbf{\Pi}_1(\boldsymbol{B}_1)]$ stands for the *OF* values at the vertex projections. Fig. 1 illustrates this computation.

In the computation of $L_{OF}^{1\to2}$, the two images play asymmetric roles. To remedy this, we also compute $L_{OF}^{2\to1}$ by reversing the roles of the images and take our full loss to be

$$L_{OF} = \frac{1}{2}(L_{OF}^{1\to2} + L_{OF}^{2\to1}) \,. \tag{2}$$

In other words, we compute the flow in both directions and average the corresponding losses. This contributes to stabilizing the training for two reasons. First, by getting *OF* information in both directions, we effectively average the estimates and reduce the noise. Similarly, by using two distinct body estimates $\boldsymbol{B}_1$ and $\boldsymbol{B}_2$, we also reduce the influence of slight errors in these. Later in Sec. 5, we discuss the technical design of the $L_{OF}$ term of Eq. 2 and the pitfalls to be avoided when implementing it.

**$OF$ Alignment as Weak Labelling.** A key feature of our approach is that we use video sequences at training time but we don't need them at inference time. This raises the question of why this additional optical flow supervision helps. Our interpretation is that minimizing $L_{OF}$ aligns the predictions of the network *BL* with the motion, as illustrated by Fig. 2. The subtraction in Eq. 1 can be rewritten as follows. For any vertex $v$ with projections $p_1$ and $p_2$ in images $\mathbf{I}_1$ and $\mathbf{I}_2$, we have

$$\begin{aligned} \mathbf{F}_{B} - \mathbf{F}_{OF}[v] &= (p_2 - p_1) - OF[p_1] \\ &= p_2 - (p_1 + OF[p_1]) \\ &= p_2 - \tilde{p}_2 \end{aligned} \tag{3}$$

with $\tilde{p}_2 = p_1 + OF[p_1]$. Thus, $\tilde{p}_2$ can be viewed as a weak label for the $p_2$ estimate. In other words, motion alignment essentially serves as weak labelling process and Eq. 2 can be interpreted as a point-to-point loss when the inference is performed for image pairs.

**Anchoring Strategy.** Given our pre-trained network *BL*, simply minimizing $L_{OF}$ of Eq. 2 tends to encourage the network to shrink the area of its projected predictions. In theory, the loss function can be brought to zero by covering a small region with constant flow, or even zero flow, with the body in image $\mathbf{I}_1$ and performing a translation, or no motion at all, to $\mathbf{I}_2$.

Hence, during fine-tuning, we do not want to deviate too much from the initial state of the pre-trained model. Assuming that the data used for pre-training is still available, we can take the full loss to be

$$L = L_{sup} + L_{OF} \,, \tag{4}$$

where $L_{sup}$ is the loss originally used to train the network. This is simple but effective. In the supplementary material we discuss alternative approaches that can be used when the original training data is unavailable.

### 3.3 Providing a Temporal Context

State-of-the-art video-based methods use special-purpose layers—GRUs, LSTMs or transformers—to account for temporal consistency. Hence, they require image sequences *both* for training and testing. By contrast, we distill the knowledge from unlabeled videos during training, regardless of whether such information is available for testing. What is more, if a full video sequence is available at test time, we can exploit it as follows.

In practice, the *BL* of Sec. 3.1 relies on a ResNet (He et al., 2016) feature encoder connected to a SMPL decoder. We exploit this by introducing a feature-wise affine transformation conditioned on the predictions for the previous frames, inspired by (Perez et al., 2018). Let us assume we are given $N$ consecutive frames. We use our *BL* to produce SMPL estimates for $N-1$ of them, $\left[\{\boldsymbol{\theta}_1, \boldsymbol{\beta}_1, \boldsymbol{\pi}_1\}, ..., \{\boldsymbol{\theta}_{N-1}, \boldsymbol{\beta}_{N-1}, \boldsymbol{\pi}_{N-1}\}\right]$. We feed these $N-1$ estimates to an auxiliary lightweight 2-MLP context network that outputs the parameters $\{\boldsymbol{\gamma}, \boldsymbol{\delta}\}$ of an affine transformation that is applied to the activation $\mathbf{f}$ of the last linear layer of the ResNet feature encoder. We write

$$\hat{\mathbf{f}} = \boldsymbol{\gamma} \odot \mathbf{f} + \boldsymbol{\delta} \tag{5}$$

and feed the $N$th frame to the network thus modified.

In other words, we still use the same single-frame network as before but inject temporal context via feature normalization. In effect, the MLP provides a temporal context for pose estimation in frame $N$. The training as described in the previous section is mostly unchanged. The only difference is that the auxiliary "context" network is now used to estimate $\boldsymbol{B}_N$ during training and learns cues from the previous estimates. Once trained, we use it at inference time as well. We provide more details in the supplementary material.

## 4 Experiments

In this section, we first discuss our experimental setup (Sec. 4.1) and demonstrate the superiority of our approach over state-of-the-art competing method (TexturePose) alone, along with the complementarity of the two approaches given varying amounts of annotations (Sec. 4.2). Next, we showcase our ability to exploit sets of unannotated video sequences at training time (Sec. 4.3) and to exploit temporal consistency not only at training time but also at test time (Sec. 4.5). We then show that our approach boosts performance regardless of backbone architecture (Sec. 4.6). Finally, we analyze the quality of *OF* as a supervisory signal (Sec. 4.7).

### 4.1 Experimental Setup

We use a well established ResNet-based architecture (Kanazawa et al., 2018; Kolotouros et al., 2019; Joo et al., 2020; Kocabas et al., 2020) as the *BL* baseline of Sec. 3.1 for most of the experiments unless stated otherwise. More sophisticated architectures, such as the volumetric heatmap-based models (Li et al., 2021) or the transformer-based HMR 2.0 architecture (Goel et al., 2023) deliver state-of-the-art results in human pose and shape estimation. However, they are very data-demanding and do not work properly under only limited supervision (Sun et al., 2018; Iskakov et al., 2019; Dosovitskiy et al., 2020). Hence, they are less suitable for our main experiments in the low-data regime, yet we demonstrate that our method can be applied to more sophisticated architectures, such as HRNet-W32 (Sun et al., 2019; Kocabas et al., 2021). To estimate optical flow we use the off-the-shelf pre-trained RAFT (Teed & Deng, 2020), without fine-tuning it.

**Pre-Training and Datasets.** To pre-train our baseline network *BL*, we use subsets obtained by randomly picking 10%, 20%, 50% and 100% of the samples from the COCO (Lin et al., 2014) dataset (only training subset, 75K) of in-the-wild images with high-quality pseudo-ground-truth supervision acquired by EFT (Joo

et al., 2020). We will refer to them as COCO$^{p\%}$, where $p$ is the percentage of the COCO images and corresponding ground-truth labels used to pre-train the baseline model.

The COCO-EFT dataset exhibits many good features. First, it is a compact yet high-quality dataset that is sufficient to train a strong body estimator model approaching state-of-the-art performance. Even when only a small fraction of the data is used (COCO$^{10\%}$, COCO$^{20\%}$), the quality of the trained baseline is satisfactory and serves as a good starting state for further refinement.

For datasets containing videos, we also use 3DPW (von Marcard et al., 2018) (both training and testing subsets), Human3.6M (Ionescu et al., 2014), and PennAction (Zhang et al., 2013) (we will refer to it as PA).

These datasets differ in size, image resolution, and scene complexity (samples from each of the datasets can be found in the supplementary material). During the fine-tuning experiments, we utilize only individual images without reference to the additional data provided in these datasets. For evaluation, we use common pose benchmarks, such as 3DPW (von Marcard et al., 2018), Human3.6M (Ionescu et al., 2014) and MPI-INF-3DHP (Mehta et al., 2017). We rely on the Adam (Kingma & Ba, 2015) optimizer, and the loss weights were chosen via grid search. We provide the details in the supplementary material.

**Metrics.** We adopt the widely used 3D Mean-Per-Joint Position Error with Procrustes Alignment (P-MPJPE, $mm$). To compare against video-based methods (Luo et al., 2020; Choi et al., 2021; Kocabas et al., 2020), we estimate the pose in each image individually and, even though our method does not feature a temporal module, we compute the acceleration error (Accel.Err, $mm/s^2$) between the predicted and ground-truth 3D joints. This metric has been used in previous works (Kocabas et al., 2020; Kanazawa et al., 2019; Choi et al., 2021) to quantify the trade-off between 3D pose accuracy and motion consistency.

## 4.2 Fine-tuning with Optical Flow Loss

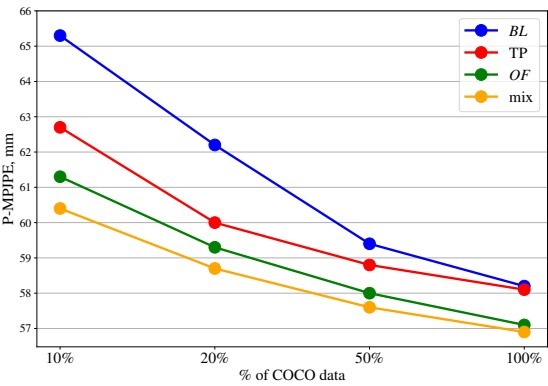

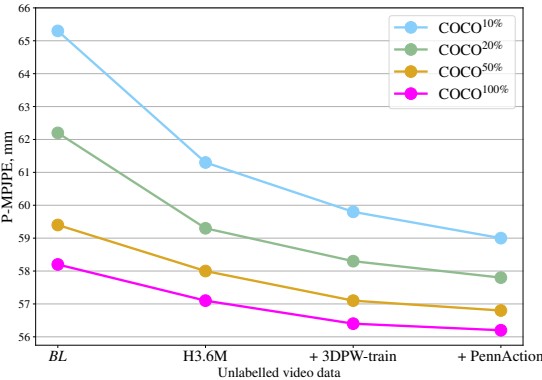

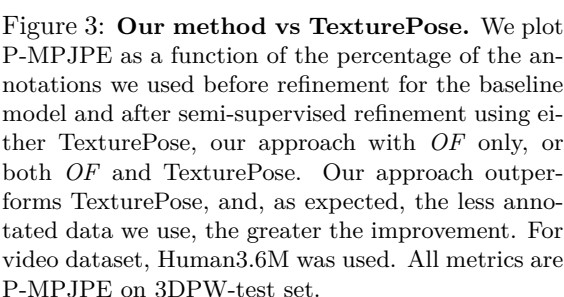

Figure 3: **Our method vs TexturePose.** We plot P-MPJPE as a function of the percentage of the annotations we used before refinement for the baseline model and after semi-supervised refinement using either TexturePose, our approach with *OF* only, or both *OF* and TexturePose. Our approach outperforms TexturePose, and, as expected, the less annotated data we use, the greater the improvement. For video dataset, Human3.6M was used. All metrics are P-MPJPE on 3DPW-test set.

Figure 4: **More unlabelled videos.** We plot P-MPJPE as a function of the amount of unlabelled data used for fine-tuning *BL* using our approach. As expected, the more unannotated data we use, the greater the improvement. All metrics are P-MPJPE on 3DPW-test set.

Table 1: **Fine-tuning the baseline model** **BL** using our *OF*-guided method (*OF*), TexturePose (Pavlakos et al., 2019) (TP), 2D keypoints (2D) (Wu et al., 2019) or mixing *OF* with each of these. For video dataset, Human3.6M was used. All metrics are P-MPJPE on 3DPW-test set.

|  | 10% | 20% | 50% | 100% |
|---|---|---|---|---|
| *BL* | 65.3 | 62.2 | 59.4 | 58.2 |
| TP | 62.7 | 60.0 | 58.8 | 58.1 |
| *OF* | 61.3 | 59.3 | 58.0 | 57.1 |
| TP+*OF* | 60.4 | 58.7 | 57.6 | 56.9 |
| 2D | 58.4 | 57.1 | 56.4 | 56.0 |
| 2D+*OF* | 57.6 | 56.5 | 56.0 | 55.8 |

Table 2: **More unannotated data.** Starting from the *BL* pre-trained using $COCO^{100\%}$, we refine it using images from either H36M alone, or H36M and 3DPW-tr, or H36M, 3DPW-tr, and PA. We evaluate on common pose benchmarks. As expected, the more images we use, the better the performance. The last row of 3DPW-test evaluations corresponds to using the test-set itself for supervision, which is meaningful and practical in some cases because the loss is self-supervised. This yields the best performance but using H36M + 3DPW-tr + PA comes close. All metrics are P-MPJPE, except Accel.Err for 3DPW-test evaluation (in brackets).

|  | 3DPW-test | H36M (P2) | MPI-INF-3DHP |
|---|---|---|---|
| *BL* ($COCO^{100\%}$) | 58.2 (29.0) | 64.4 | 67.3 |
| H36M | 57.1 (24.4) | 59.7 | 65.2 |
| H36M + 3DPW-tr | 56.4 (24.0) | 59.4 | 65.0 |
| H36M + 3DPW-tr + PA | 56.2 (23.9) | 59.1 | 64.7 |
| 3DPW-test | *55.8 (20.2)* |  |  |

We compare our approach against TexturePose (Pavlakos et al., 2019), the only other self-supervised pose and shape estimation method we know of that relies mostly on image content as we do. Along with color-consistency, TexturePose exploits silhouettes and 2D keypoints, which we do not.

We first use the ground-truth body poses of the $COCO^{p\%}$ datasets to pre-train *BL*. We then use the video sequences of Human3.6M (Ionescu et al., 2014) and only them—-we ignore the ground-truth pose data—to refine the model weights by minimizing the loss of Eq. 4 that includes the optical flow loss of $L_{OF}$ of Eq. 2 or, instead, one that includes the texture-aware loss term of TexturePose. To this end, we re-implemented the texture extraction algorithm of TexturePose because the original code is not publicly available. This has enabled us to compute the loss

$$L = L_{sup} + L_{\text{TP}} \ , \tag{6}$$

where $L_{\text{TP}}$ is the texture-aware loss of TexturePose, that plays the same role as $L_{OF}$ in our formulation. Our training batches are combinations of independent frames with single-frame ground-truth supervision and pairs of consecutive frames without annotations. For TexturePose, we sample 5-frame sequences as in the original paper (Pavlakos et al., 2019).

We evaluate both approaches on the test set of the 3DPW dataset, which was not used in any part of the training. In Fig. 3, we plot the P-MPJPE metric as a function of what percentage of the annotated data we used to pre-train the pose estimation baseline *BL*. Using either method, the less initial supervision is used initially the more noticeable the improvement brought about by the unannotated sequences. However, our approach consistently outperforms TexturePose and delivers an improvement even when the baseline has been trained using *all* the available annotations, whereas TexturePose does not. We attribute this to the fact that our approach directly leverages all the 3D points of the body mesh, whereas TexturePose works by interpolation in the pixel space. Note that the model that uses only 10% of the available supervision coupled with the proposed *OF* loss beats the baseline model that uses twice as many samples. The same can be seen in Fig. 3 when comparing models of *OF*(50%) and *BL*(100%).

To demonstrate that the two approaches are in fact complementary, we repeat the experiment by minimizing a loss that combines ours and that of TexturePose. We take it to be

$$L = L_{sup} + L_{OF} + L_{\text{TP}} \ , \tag{7}$$

where $L_{sup}$ and $L_{OF}$ are the losses of Eq. 4, while $L_{\text{TP}}$ is the texture-aware loss of TexturePose. As can be seen in Fig. 3, this yields a further improvement over optical flow alone, especially when using small fractions of the available annotations to pre-train the network. Our interpretation is that minimizing the $L_{\text{TP}}$ term helps correct the body estimates so that more of its projections overlap the true body, which ensures that the flow is used more effectively.

To put these numbers in context, the best current method (Goel et al., 2023) delivers a P-MPJPE of 44.4 but relies on vast amounts of training data and a transformer architecture that is unlikely to do well with

reduced amount of training of data. Similarly, the method of (Li et al., 2021) delivers 48.8 but heavily relies on a strong 3D heatmap estimator that also requires abundant supervision.

### 4.3 Using more Unannotated Data

The results shown above were obtained using Human3.6M as the sole source of unlabeled videos. Since we have additional datasets, we can use them as additional source of unannotated images. We ran this experiment and report the results in Table 2 (across various evaluation benchmarks) and in Fig. 4 (across baseline models that use different amount of supervision). We started from the baseline models $BL$ pre-trained with subsets of COCO and refined using only Human3.6M (Ionescu et al., 2014) as before, or adding images from 3DPW-train (von Marcard et al., 2018), or from both 3DPW-train and PennAction (Zhang et al., 2013). As expected, the more unannotated images we use, the better the result across all benchmarks.

Furthermore, because our method is semi-supervised, we can use the test data to further refine the network. In the last row of Table 2, we report the result of using the test videos to refine the $BL(\text{COCO}^{100\%})$ network. This yields the best results of all and there are scenarios in which this would be practical. However, when using all the available data but not the test data, we obtain similar results.

As before, the less supervision the model has received, the more valuable unsupervised $OF$ motion alignment becomes (-6mm in P-MPJPE for $\text{COCO}^{10\%}$). The baseline model trained with $\text{COCO}^{50\%}$ is considerably outperformed by the fine-tuned one that has seen 5 times less supervision ($\text{COCO}^{10\%}$) but used the proposed weakly-supervised $OF$-induced motion alignment.

To put these results in context in terms of P-MPJPE (mm), our model that utilized 20% of available body supervision $OF(\text{COCO}^{20\%})$ model (57.8) outperforms such models, as SPIN (Kolotouros et al., 2019) (59.2), PyMAF (Zhang et al., 2021) (58.9) and performs on par with I2L-MeshNet (Moon & Lee, 2020) (57.7). All these methods rely on full body supervision using millions of samples, whereas our approach uses only about 15K.

Although additional data brings a significant boost to the models' performance, it comes at cost of longer training time. We provide these statistics in the supplementary material.

### 4.4 Beyond Unannotated Videos

In keeping with the central focus of this paper, so far, we have only shown results using unlabeled videos to provide additional supervision. This restricts the range of methods against which we can compare. To broaden it, we now use the Detectron2 2D keypoint estimator (Wu et al., 2019) as an alternative baseline. Although training Detectron2 requires a large amount of body-related data, which runs against our desire to use less supervision, we can treat it as a black-box model that provides additional supervision and run it on our video frames during training. In addition, one can also combine our optical flow approach with this additional 2D keypoint supervision to show their complementarity.

More specifically, we take the loss to be the combination of the fully supervised $L_{sup}$ and the additional $L_{2D}$ alignment between 2D keypoints predicted by off-the-shelf 2D pose estimator and the joints extracted from the predicted body shape. As we did with TexturePose in Section 4.2, we repeat the experiment by minimizing a loss that combines ours and 2D keypoint alignment, as in Eq. 7. In the last two rows of Tab. 1, we compare 2D keypoint supervision (2D) and the combination of our method with 2D keypoints (2D + $OF$). Even though 2D keypoints encode more body-related information, optical flow provides useful additional information and enhances the model's performance.

### 4.5 Using the Temporal Context

As discussed in Sec. 3.3, $OF$-guidance can be used to exploit temporal information in video sequences. In other words, even though one often distinguishes between single-frame and video-based methods, our approach bridges the gap between the two. It enables a single-frame model to exploit motion information at training time to provide more realistic motions at inference time.

Table 3: **Our approach with and without temporal context.** Starting from a pre-trained baseline model using $COCO^{100\%}$, we report results using from 0 to 8 previous frames. Here 0 stands for not using the temporal context and operating strictly on one single frame, as in Sec. 4.2. We also compare against state-of-the-art video-based methods. Our method performs on par with them in both error in pose and, most importantly, error in acceleration. In other words, it provides less shaky estimates, even though it was not tailored for video-based estimation. All metrics are P-MPJPE on 3DPW-test set.

| | | P-MPJPE | Accel.Err |
|---|---|---|---|
| ctx length | 0 | 57.1 | 24.4 |
| | 1 | 57.0 | 24.0 |
| | 2 | 56.9 | 22.6 |
| | 4 | 56.7 | 19.4 |
| | 8 | 56.7 | 18.8 |
| *video-based* | | | |
| HMMR (Kanazawa et al., 2019) | | 72.6 | 15.2 |
| VIBE (Kocabas et al., 2020) | | 57.6 | 25.4 |
| MEVA (Luo et al., 2020) | | 54.7 | 11.6 |
| TCMR (Choi et al., 2021) | | 52.7 | 6.8 |
| MPS-Net (Wei et al., 2022) | | 52.1 | 7.4 |

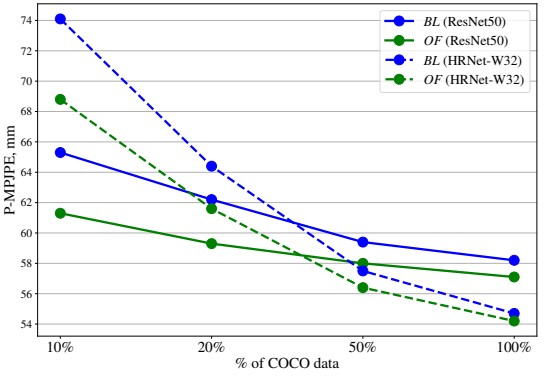

Figure 5: **Different backbones.** We apply our fine-tuning strategy using two different backbones, ResNet50 (*solid*) and HRNet-W32 (*dashed*). Our approach delivers a significant improvement in both cases. Note that the complex volumetric features of HRNet-W32 perform better than those of ResNet when we use a lot training data and worse when we do not.

Figure 6: **OF quality.** We compute the distances $d_{OF}$ and $d_B$ as described in Sec. 4.7 for a particular video sequence. We see that the motion flow induced by the body estimator lies always further from the ground-truth, than the optical flow estimate. It proves the effectiveness of the off-the-shelf optical flow model.

To demonstrate this, we use the same experimental setup as in Sec. 4.2. We compare the performance of the network pre-trained using $COCO^{100\%}$ and refined using the Human3.6M sequences, with the same network trained to use the predictions obtained in the previous $N-1$ images, with $N$ ranging from 1 to 8. We report the results in Table 3. Even a very short 2-frame time window gives a small P-MPJPE boost. It increases slightly for $N = 8$ but not for greater values of $N$. More importantly, exploiting temporal consistency provides a more significant boost in the Acceleration Error introduced in Sec. 4.1, meaning that the sequence of recovered poses will be far less jerky, even though it ls produced by a single-frame inference model without recurrent layers. We would like to emphasize that prediction-from-video is not our primary focus. Instead, we demonstrate how to exploit temporal information that is naturally encoded in the optical flow, and how it makes the single-frame model compatible with the video inputs. More experiments involving a temporal context are provided in the supplementary material.

Table 4: **OF quality.** The mean and median of the ratios between $d(\mathbf{F}_B, \mathbf{F}_{\mathrm{GT}})$ and $d(\mathbf{F}_{OF}, \mathbf{F}_{\mathrm{GT}})$ across 1000 random samples from each dataset, as described in Sec. 4.7. $\Delta t$ is a time distance between two frames in a sequence. *Oracle* refers to the case where *OF* is computed at ground-truth vertices instead of estimates of the body regressor.

| $\Delta t$ | 3DPW | | H36M | | PennAction | |
|---|---|---|---|---|---|---|
| | LR | HR | LR | HR | LR | HR |
| 1 | 2.3 / 1.6 | 2.6 / 1.8 | 2.4 / 1.7 | 2.7 / 1.9 | 3.2 / 1.9 | 3.3 / 1.9 |
| 3 | 1.6 / 1.3 | 1.9 / 1.3 | 1.6 / 1.3 | 1.8 / 1.5 | 2.8 / 1.4 | 2.3 / 1.4 |
| 5 | 1.4 / 1.1 | 1.5 / 1.2 | 1.5 / 1.2 | 1.6 / 1.3 | 2.7 / 1.4 | 2.1 / 1.4 |
| 7 | 1.3 / 1.1 | 1.4 / 1.2 | 1.4 / 1.2 | 1.5 / 1.3 | 2.2 / 1.3 | 2.5 / 1.3 |
| 1 (*oracle*) | *6.4 / 4.0* | *6.7 / 4.1* | *7.9 / 5.0* | *8.8 / 4.9* | *6.6 / 3.2* | *7.0 / 3.1* |

## 4.6 Changing the Backbone Architecture

To illustrate that our method improves performance regardless of backbone architecture, we replace ResNet-50, which we have used so far, with the high-resolution backbone HRNet-W32. It was first proposed in (Sun et al., 2019) and later adapted for SMPL estimation in (Kocabas et al., 2021). We pre-train (*BL*) and fine-tune (*OF*) this new architecture as in Sec. 4.2. We report our results in Fig. 5. HRNet-W32 learns high-resolution volumetric features. This is very effective when there is enough training data and HRNet-W32 provides state-of-the-art result when using *all* the training data. However, in the data-scarce regime, the volumetric features are much harder to learn (Sun et al., 2018; Iskakov et al., 2019). Hence, under these conditions, ResNet delivers better results.

In any event, our proposed fine-tuning strategy improves performance considerably, as it did in the case of the ResNet backbone presented in Sec. 4.2. In both cases, the less data is used for supervision, the more noticeable the improvement is. In other words, our method is model-agnostic and is beneficial independently of the specific network architecture being used.

## 4.7 Quality of *OF* as a Supervisory Signal

In this section, we demonstrate that the *OF*-predicted motion is far better aligned with the real motion of the body than the motion that can be inferred from the pre-trained baseline, which goes a long way towards explaining why it provides a useful supervisory signal.

To this end, we exploit the fact that our datasets feature 2D ground-truth keypoints. For any given pair of frames $\mathbf{I}_1$ and $\mathbf{I}_2$ we compute three motion flows at the positions of visible keypoints:

1. from the ground-truth 2D keypoints, $\mathbf{F}_{\mathrm{GT}}$,

2. from body estimates, $\mathbf{F}_B$,

3. and the optical flow one, $\mathbf{F}_{OF}$.

Then, we compute $d_{OF} = d(\mathbf{F}_{OF}, \mathbf{F}_{\mathrm{GT}})$ and $d_B = d(\mathbf{F}_B, \mathbf{F}_{\mathrm{GT}})$, where $d$ is the usual Euclidean distance. In Fig. 6, we plot these two distances—$d_{OF}$ in blue and $d_B$ in red—for a specific sequence. Note that the blue curve is systematically below the red one, indicating the optical flow estimate is closer than the body regressor estimate to the ground-truth.

To show that this is true not only for a single sequence but in general, we generated Table 4. The two numbers in each cell are the mean and the median of the ratios of the two distances for a randomly chosen set of 1000 images from each dataset. The higher these numbers the more additional information *OF* is likely to provide. Since image resolution strongly affects the quality of the *OF*, for each image, we compute *OF* map at two resolutions, low (224 pixels, *BL* input format) and high (maximum available). Each line in the table corresponds to a different time interval between frames in which we compute the *OF*. Since the predicted bodies do not project perfectly on the image, the *OF* can be corrupted by background pixels. To gauge this effect, we recomputed $d_{OF}$ using the ground-truth body models to ensure that all pixels involved

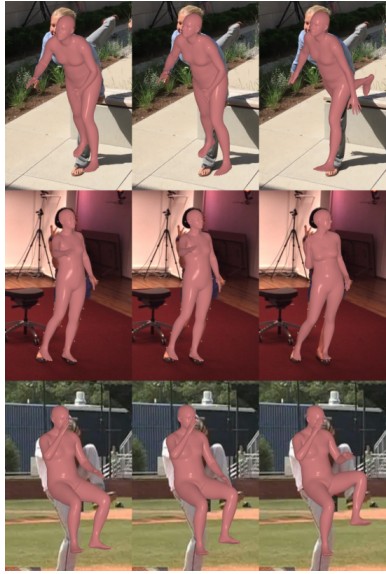

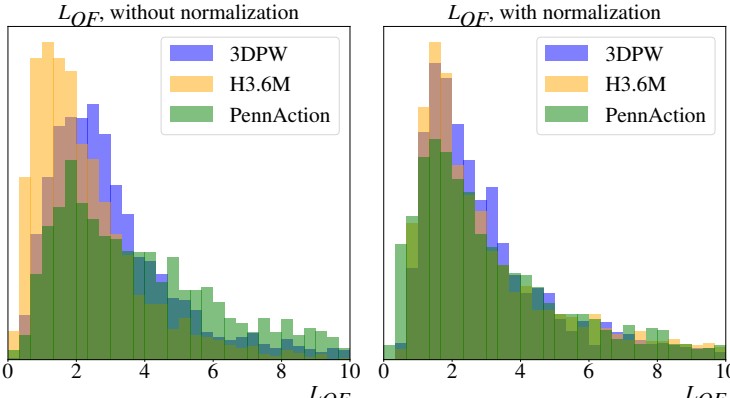

Figure 7: **Randomization.** The first two columns in every row show pairs of consecutive frames from different video datasets, 3DPW, H36M and PennAction, with overlaid body estimates. These estimates are very similar to each other but wrong. The third column feature the body estimate for the second frame after color randomization. Even if this does not give the correct motion flow or $OF$, it disentangles body estimates from each other.

Figure 8: **Scaling of $OF$ loss.** We found that the differences in motion behavior may drastically differ across different datasets or even same video sequences. On the *left*, we show the values of $L_{OF}$ (Eq. 2) for $2k$ samples of each of the video datasets. We propose to normalize $L_{OF}$ for every sample over the mean of the $OF$ norms across all visible vertices. The result is on the *right*. Loss normalization significantly stabilizes the training.

are foreground pixels. This can be viewed as an oracle and is given on the final line of the table. As the training progresses and the predictions improve, the ratios can be expected to approach these values. In any event, these results confirm our hypothesis that $OF$ brings valuable additional information and that $\Delta = 1$, i.e. using consecutive frames, is the best way to exploit it.

## 5 Implementation details

In this section, we discuss the technical design of the $L_{OF}$ from Eq. 2 and what pitfalls should be avoided when implementing it. In particular, we added color noise to differentiate the predictions of two neighboring frames, threshold and scale the loss values for computational stability.

**Rigidity of body estimators.** For the experiment carried out in Sec. 4.2, we found that the $L_{OF}$ from Eq. 2 must be implemented carefully. Body estimates for neighboring frames, especially when acquired by a fixed camera as in Human3.6M (Ionescu et al., 2014), look very much alike, because the images and the features extracted from them are also similar. Consequently, similar frames can yield similarly bad body estimates, see the first two columns in Fig. 7 for an illustration. Hence, the seemingly natural assumption that body estimates for different inputs are independent does not hold, which can lead to *correlated* errors in $OF$ estimates of Eq. 1. We can eliminate this effect by adding random noise to the colors of input frames. This makes the body estimation model more robust to perturbations in pixel colors and allows it provide different outputs. See the third column of Fig. 7.

**Thresholding of $OF$ loss.** Additionally, for computational stability, very large values of $L_{OF}$ should be discarded. Our tests show that such high spikes in $OF$ estimates are connected to either wrongly chosen

pairs of frames—in some sequences, some frames were cut out—or to partial occlusions. We threshold $L_{OF}$ by the maximum of $OF$ norms $\|\mathbf{F}_{OF}\|$ across all visible vertices.

**Scaling of $OF$ loss.** When using different video datasets for refinement, the motion characteristics, such as default FPS, can be very different and very different behaviors can be seen even among the pairs of frames inside one video sequence. For example, in Human3.6M (Ionescu et al., 2014) in most of the videos, the subject stands still in the first several frames before starting to move to perform actions. In such cases, the estimated motion from both optical flow or body pose&shape predictor may differ in orders of magnitude, as shown in Fig. 8 (*left*). We found it crucial to compensate for this effect when minimizing the loss of Eq. 4, because it has a supervised part that is independent from the motion changes. To this end, for a given body estimate, we compute the mean value of the $OF$ norms $\|\mathbf{F}_{OF}\|$ across all visible vertices, and divide the value of $L_{OF}$ over it. In Fig. 8, we plot histograms of $L_{OF}$ values before fine-tuning the baseline model. Every histogram contains 2000 datapoints randomly sampled from each of the datasets. The plot on the left shows how diverse and inhomogeneous the motion is. The plot on the right shows the same values after normalization. Now all datasamples are equally distributed.

## 6 Conclusion

We have shown that optical flow between image pairs could be used to refine a pre-trained network so that it works better on *single images*. This can be ascribed to the fact that the motion estimates delivered by a good but standard $OF$ estimator correspond better to the real motion than what can be inferred from the pre-trained pose and shape estimator's predictions. This is a useful observation because all it takes to exploit are video sequences of people in motion, which are easy to acquire. Thus, even though we acknowledge the prevailing "data is king" paradigm, such scenarios do occur due to privacy and licensing issues, which makes this useful. Furthermore, our method also aligns with ethical considerations, while respecting data minimization principles.

In principle, our optical flow-based technique could be applied for other moving objects. We plan to extend our approach to a very challenging domain of animal pose and shape estimation. There is an abundant amount of unannotated videos, yet few methods and models can exploit them effectively because there is very little annotated data for this task. We will also exploit motion priors that can be obtained from existing databases to further constrain the refinement.

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

Table 5: Overview of the video datasets used in this work with the number of images and image resolution.

| Datasets | #samples | Image resolution |
|---|---|---|
| 3DPW-train | 23k | 1080x1920 px |
| 3DPW-test | 34k | 1080x1920 px |
| Human3.6M | 293k | 1002x1000 px |
| PennAction | 162k | 360x480 px |

Table 6: **Using more unannotated data from 3DPW-train and PA.** Despite PA being much larger (x7) than 3DPW-train, the improvement coming from it is smaller. We explain it with the low image resolution of PA (and hence, the lower quality of *OF*) and the larger domain gap with the 3DPW-test set used for evaluation.

| | P-MPJPE ($\downarrow$) | Accel.Err ($\downarrow$) |
|---|---|---|
| $BL$ (COCO$^{100\%}$) | 58.2 | 29.0 |
| PA | 57.8 | 27.8 |
| 3DPW-train | 57.4 | 25.8 |
| 3DPW-tr + PA | 57.2 | 25.2 |

In this supplementary material, we provide the quality of the estimated motions with the corresponding videos attached (Sec. A), the choice of hyper-parameters (Sec. B), the samples from each of the datasets with the pre-processing details (Sec. C), additional experiments on using more unlabelled data with combinations of the datasets not used in the main paper (Sec. D), the architectural design of the temporal context injection (Sec. E), additional experiments on using temporal context for baselines of less supervision (Sec. F) and alternative strategies of exploiting optical flow loss, when the supervision is not available (Sec. G). The code and supplementary files are publicly available at https://github.com/cvlab-epfl/of4hmr.

## A  Motion quality analysis

Here we evaluate the estimated motions on particular sequences qualitatively and quantitatively. In Fig. 9, we plot acceleration errors for four video sequences and compare the baseline trained on COCO$^{100\%}$ with its *OF*-refined versions without and with the temporal context (lines "0" and "8" in Table 3 in the main paper). It is clear that the proposed *OF* strategy improves the estimated motion considerably.

Furthermore, we provide video files with the corresponding body renderings for qualitative comparison.

## B  Optimization details

In all our experiments in the main paper, we used grid search to find the optimal balance between different terms in loss functions. The values for the balance weights are: $\lambda[L_{sup}] = 1$, $\lambda[L_{OF}] = 0.01$, $\lambda[L_{\text{TP}}] = 10$.

## C  Datasets

We provide samples from each of the datasets used in the present paper in Fig. 10.

**Pre-processing of video datasets.**  We follow the common pre-processing of video datasets, first proposed in (Kocabas et al., 2020). Their pre-processed dictionaries contain lists of frames grouped by videos. We provide the number of samples and image resolution of the video datasets in Table 5. During our experiments, we did not use any image augmentation on video frames, except color noise, when inferring body estimates (the details are in Sec. 5).

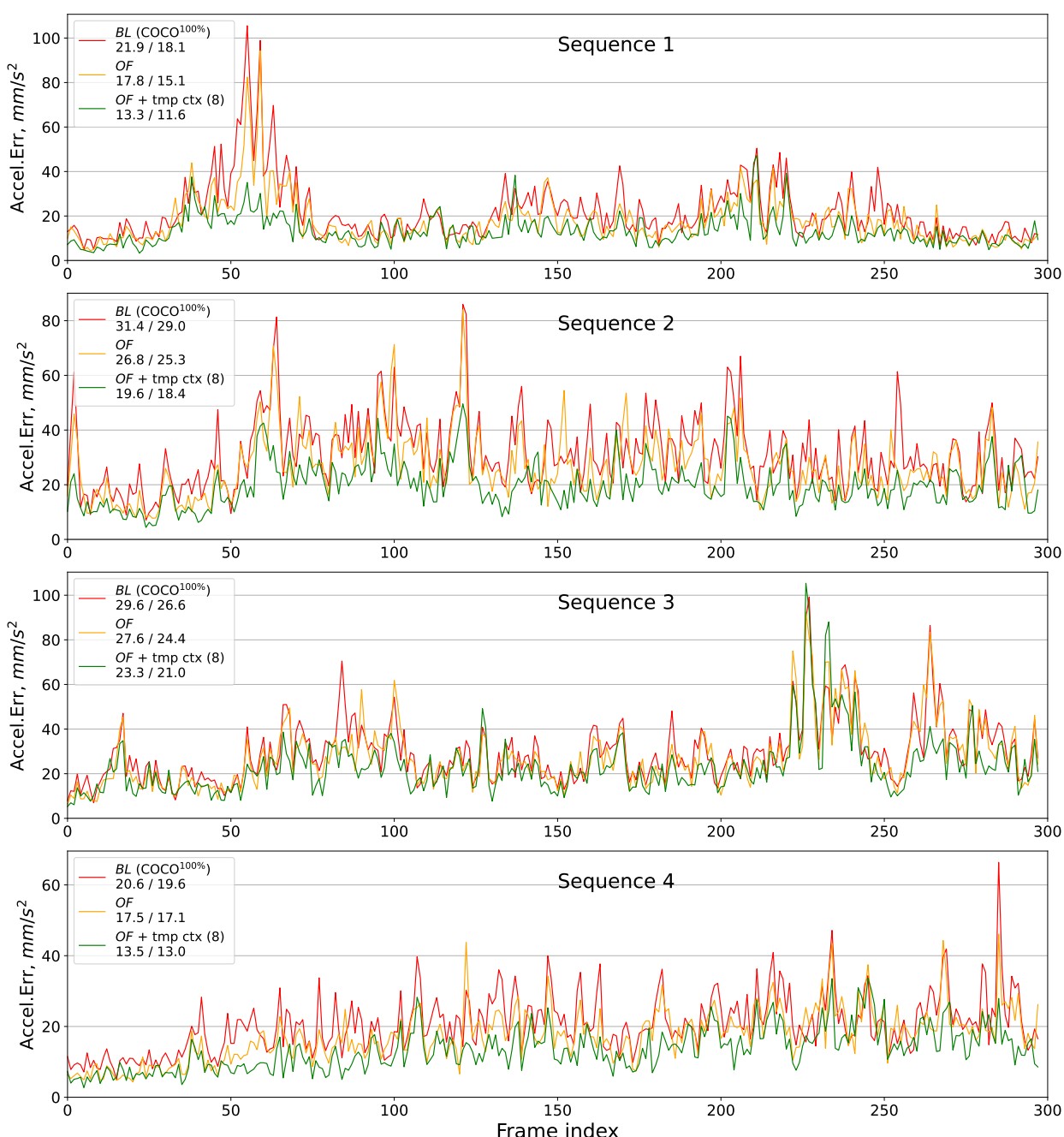

Figure 9: We plot Accel.Err for four particuluar video sequences. Colored lines show the performance of the baseline model *BL* (COCO$^{100\%}$), its *OF*-refined version (Sec. 4.2 of the main paper) and *OF*-refined with temporal extension (Sec. 4.5 of the main paper). In the legends, we report the mean and the median across all frames in a sequence. It is evident that the proposed *OF*-based semi-supervised technique and its temporal context version significantly improve human motion, despite the fact that the architecture is single-frame and does not contain any recurrent layers. We provide corresponding video files with side-by-side motion comparison.

# D   Additional results on "Using more Unannotated Data"

In Table 2 of the main paper, we presented how adding more unannotated data brings more improve-ment to the performance of the body estimator pretrained using COCO$^{100\%}$. In the main paper, we took

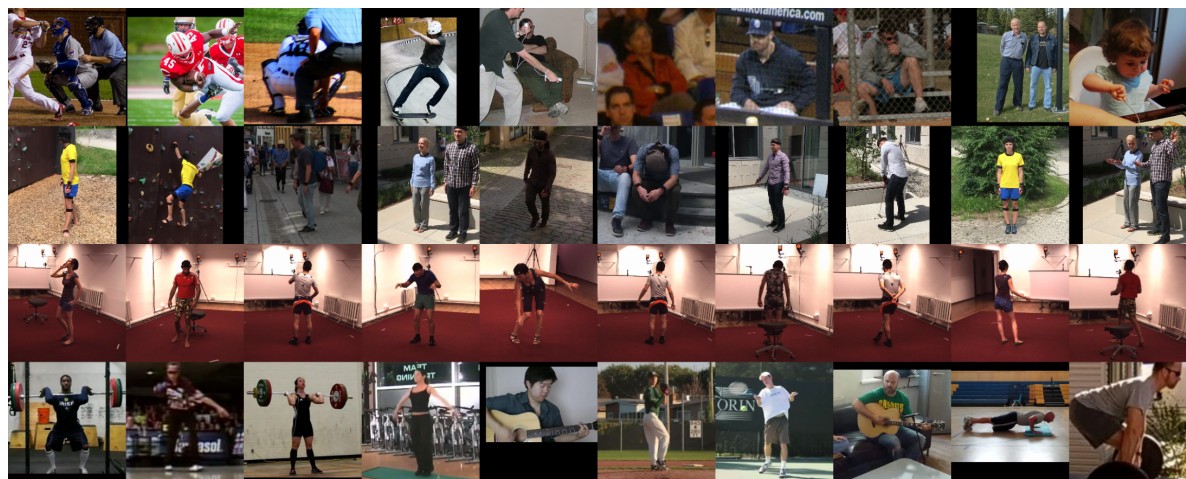

Figure 10: **Samples from the datasets exploited in this work.** From top to bottom: COCO (Lin et al., 2014), 3DPW (von Marcard et al., 2018), H36M (Ionescu et al., 2014) and PennAction (Zhang et al., 2013). COCO and its extended version from (Joo et al., 2020) provides high-quality pseudo ground-truth bodies for individual frames. All other datasets comprise video sequences that we use without labels.

Table 7: **The cost of using more data.** The use of more data comes with larger cost of training time. The "#samples" column denotes the number of additional training samples, and the "#iterations" shows the number of the network weights updates needed for model convergence during fine-tuning.

|  | P-MPJPE ($\downarrow$) | Accel.Err ($\downarrow$) | #samples | Training time, #iterations |
|---|---|---|---|---|
| H36M | 57.1 | 24.4 | 293k | 150k |
| H36M + 3DPW-tr | 56.4 | 24.0 | 316k | 160k |
| H36M + 3DPW-tr + PA | 56.2 | 23.9 | 478k | 270k |
| 3DPW-test | 55.8 | 20.2 | 34k | 30k |

Human3.6M (Ionescu et al., 2014) as the starting set and then extended it. Here we present additional experiments with combinations of 3DPW-train (von Marcard et al., 2018) and PennAction (PA) (Zhang et al., 2013), see Table 6.

We note in the main paper that *OF* quality depends on the image resolution. The relatively low resolution of PennAction (Zhang et al., 2013) images (see Table 5) partially explains the low performance of fine-tuning using these frames, despite the fact that the amount of samples is several times larger than 3DPW. Another reason is the large domain gap between the 3DPW and PennAction images (as can be seen in Fig. 10). By contrast, the gap between the training and testing sets of 3DPW is minimal, which explains why 3DPW-train outperforms PA at refinement.

In Sec. 4.3 of the main paper, we speculate that using more data requires larger resources during training. We provide the statistics on the number of samples and the training time in Table 7 for the 3DPW-test benchmark evaluation experiments, discussed in Table 2 of the main paper. The "#samples" column denotes the number of additional training samples, and the "#iterations" shows the number of the network weights updates needed for model convergence during fine-tuning.

## E  Temporal context network

In Sec. 3.3 of the main paper, we present the Temporal Context network. It takes body regressor's predictions of a sequence of $N - 1$ frames as input and outputs the parameters of an affine transformation $\boldsymbol{\gamma}$ and $\boldsymbol{\delta}$ for the $N$th frame. Every prediction $\{\boldsymbol{\theta}, \boldsymbol{\beta}, \boldsymbol{\pi}\}$ is of size $d = 24 \times 3 + 10 + 3 = 85$. The sequence of predictions is concatenated, so that the size of the input tensor becomes $N - 1 \times d$ (for each sample in the batch).

Table 8: **Temporal context with less single-frame supervision.** We report P-MPJPE and Accel.Err for the experiments with the temporal extension of our method, when less single-frame supervision is used. As expected, when less supervision is used, the greater the improvement.

| ctx length | $COCO^{10\%}$ | $COCO^{20\%}$ | $COCO^{50\%}$ | $COCO^{100\%}$ |
|---|---|---|---|---|
| 0 | 61.3 / 25.8 | 59.3 / 25.4 | 58.0 / 25.0 | 57.1 / 24.4 |
| 1 | 61.1 / 24.6 | 59.1 / 24.6 | 57.9 / 24.3 | 57.0 / 24.0 |
| 2 | 60.5 / 23.1 | 58.9 / 22.9 | 57.7 / 22.7 | 56.9 / 22.6 |
| 4 | 59.6 / 21.9 | 58.4 / 21.8 | 57.6 / 19.7 | 56.7 / 19.4 |
| 8 | 59.6 / 21.5 | 58.2 / 21.5 | 57.4 / 19.2 | 56.7 / 18.8 |

Table 9: **Unsupervised NN fine-tuning.** Initial body predictions are used for consistency according to the optimization of Eq. 8. Evaluation is performed on the 3DPW-test set.

| | P-MPJPE ($\downarrow$) | Accel.Err ($\downarrow$) |
|---|---|---|
| $BL$ ($COCO^{100\%}$) | 58.2 | 29.0 |
| 3DPW-test | 57.4 | 23.9 |
| 3DPW-train | 58.1 | 27.3 |

First, the input tensor is filtered with a 1x1 convolution layer along the context axis, with an output of size $16 \times d$. This tensor is reshaped to a vector and fed to a linear layer $16 \cdot d \to 128$. This 128-size feature is mapped to $\boldsymbol{\gamma}$ and $\boldsymbol{\delta}$ by two independent linear layers $128 \to 2048$. We add 1 to the obtained $\boldsymbol{\gamma}$, so that the network learns its deviation from 0. Finally, we transform the inner activation (of size 2048) of the ResNet's backbone using $\boldsymbol{\gamma}$ and $\boldsymbol{\delta}$, according to the Eq. 5 of the main paper. For non-linearities, we use $LeakyRelu(0.2)$. We initialize the weights of the network according to a normal distribution $\mathcal{N}(0, 0.01)$. We found it crucial to keep the parameters $\boldsymbol{\gamma}$, $\boldsymbol{\delta}$ close to 1 and 0, respectively; otherwise, we found the perturbation incurred to the network's output to be too strong.

## F  Temporal context under less supervision

In Sec. 4.2 of the main paper, we have shown that the less annotated data we use to pre-train the baseline model, the greater the improvement from our method. Following this strategy, we refine the baselines of various single-frame supervision using the temporal context network from Sec. 3.3 in the main paper and Sec. E above.

The results can be found in Table 8 (it is an extended version of Table 3 in the main paper). We see that both the position error P-MPJPE and the motion alignment Accel.Err decrease with larger temporal context, and the effect is stronger when less data is used for supervision.

## G  Alternative ways of using *OF*

### G.1  Unsupervised fine-tuning

In Sec. 3.2 of the main paper, we assumed that the annotations used to pre-train our baseline were still available when refining it using the unannotated video sequences. This may not always be the case. In such a case, we propose an alternative approach to preventing the network weights from drifting too far away from their original values. Instead of minimizing the loss of Eq. 4 of the main paper, we minimize

$$L = \lambda_\theta \|\boldsymbol{\theta}^0 - \boldsymbol{\theta}\|_2 + \lambda_\beta \|\boldsymbol{\beta}^0 - \boldsymbol{\beta}\|_2 + \lambda_{OF} L_{OF} \tag{8}$$

where $\boldsymbol{\theta}$ and $\boldsymbol{\beta}$ are the SMPL shape parameters introduced in Sec. 3.1 of the main paper and the 0 subscript denotes their initial values. We report the results of this alternative approach in Table 9, where we use the 3DPW-test set for evaluation.

Table 10: Unsupervised optimization of 3DPW-test body sequences with Eq. 8. All sequences are optimized separately. Evaluation on 3DPW-test set.

|  | P-MPJPE ($\downarrow$) | Accel.Error ($\downarrow$) |
|---|---|---|
| $BL$ (COCO$^{100\%}$) | 58.2 | 29.0 |
| $OF$ | 58.0 | 18.2 |
| $OF$ + smoothing | 57.9 | 13.3 |

Since our refining method is self-supervised, we can use the test videos for training (first row of Table 9). This regime is the so-called "test-time optimization". We see an improvement in position error, yet the acceleration error is relatively high (compared with the semi-supervised fine-tuning of Sec. 4.2). This shows that the pre-trained learning model is capable of local improvements, yet the "hard" anchoring of Eq. 8 does not allow the model weights to go too far from their initial state, hence they get trapped in a local optimum.

If the unlabeled videos are taken from the training set (last row of Table 9), we see a marginal improvement in both metrics. The over-reliance on the initial weights in Eq. 8 prevents generalization to the test set. Note, however, that we already saw the significant effect from the 3DPW-train set in Table 6, when single-frame supervision was used as anchoring.

## G.2 Body sequence optimization

As the network we use to regress body pose and shape in its basic configuration takes a single frame at a time and does not contain any recurrent modules, the output body sequence is very shaky.

Instead of fine-tuning the network, one can refine the body sequence predictions directly to make the sequence realistically smoother using $OF$-guidance (Eq. 2 of the main paper). Specifically, the SMPL parameters of the estimated sequences are optimized explicitly, using the loss from Eq. 8 anchored to the initial estimates. A similar experiment was carried out in (Tung et al., 2017), yet they used multi-camera setups, and silhouette and keypoint estimators. We show that without bells and whistles, using just a well pre-trained single-frame baseline and an off-the-shelf $OF$ estimator, the results can be satisfactory. We optimized all video sequences separately, and report the average results on 3DPW-test in Table 10. It is also useful to add smoothing as a post-processing (second row in Table 10). To do that, we added an additional term to Eq. 8, pulling the parameters $\boldsymbol{\theta}$ and $\boldsymbol{\beta}$ to their moving average (we found the best window size to be 30 frames).

We found that the human motion consistency improves considerably, while keeping the position error close to the initial.

