# OpenReview forum: "Using Motion Cues to Supervise Single-frame Body Pose & Shape Estimation in Low Data Regimes"
_TMLR — Accepted by TMLR_

### Review · Reviewer_GTP2 · 2023-11-21

**Summary Of Contributions:**

This paper presents an approach to 3D human pose and shape estimation in the absence of annotated data. It is assumed that video information is available to exploit the optical flow information. A deep neural network based on the SMPL model is pre-trained on the available data to be used later to extract pseudo-labels of shape and pose from the unlabelled videos. The optical flow is then used to propagate the SMLP model prediction from a previous frame and apply a consistency loss. In addition, the available pre-training data is used as another source of supervision. The approach is mainly compared with previous work on TexturePose on the 3DPW dataset.

**Audience:**

Yes

**Broader Impact Concerns:**

No concerns.

**Claims And Evidence:**

Yes

**Requested Changes:**

- The experimental part should be further worked based on the above comments.

**Strengths And Weaknesses:**

Paper strengths

+ The paper is well written and easy to follow. The related work is complete and it is clear where the paper advances over the previous work. In addition, the method is well written and clearly explained.

+ The paper shows promising results on a standard benchmark. Without relying on labels, it can show better performance than the previous work that uses additional amount of information for training.

+ The experiment of including additional datasets shows good generalisation capabilities.


Paper Weaknesses

+ The proposed method lacks novelty. It does not present a new methodology. However, it combines existing approaches in a novel way and this at least adds a solid contribution to the paper.

+ The paper is based solely on the SMPL model. It would add value to the paper to discuss or present an ablation study where the approach works with another body pose and shape model.

+ The paper focuses mainly on comparison with TexturePose and does not include other baselines and previous work for comparison. As the direct related work for comparison is limited, more baselines could be formed to conclude on the performance of the paper.

+ Table 2 would be more helpful if it included the additional number of training images. It could also include the additional training time to have an overview of the cost of the additional performance.

+ It is important to include more evaluations than the 3DPW test set.

---

> ### Author Response · Authors · 2023-12-19
> **Authors Response to Reviewer GTP2**
>
> We thank the reviewer ${\color{green} GTP2}$ for their time reading our paper and for their suggestions. Please, find our response to each of your questions and concerns below.
>
> _____________________________________________________________
>
> * **"It is important to include more evaluations than the 3DPW test set."**
>
> We agree that it is important to provide more evaluations of our method. We have added the evaluation on the Human3.6M (following Protocol 2) and MPI-INF-3DHP ([Mehta et al., 3DV 2017](https://vcai.mpi-inf.mpg.de/3dhp-dataset/)) benchmarks. For evauation, we take the baseline model COCO(100%) and all models fine-tuned with our method with various amounts of data. In other words, we extend Table 2 of the main paper with more evaluations. Please, find the results in the table below.
>
> |                       | 3DPW-test | Human3.6M | MPI-INF-3DHP |
> |:---------------------:|:---------:|:---------:|:------------:|
> |   BL                  |    58.2   |    64.4   |     67.3     |
> | + H36M                |    56.4   |    59.7   |     65.2     |
> | + H36M + 3DPW-tr      |    56.2   |    59.4   |     65.0     |
> | + H36M + 3DPW-tr + PA |    55.8   |    59.1   |     64.7     |
>
> Every column in the table corresponds to one of the evaluation benchmarks: 3DPW-test, Human3.6M and MPI-INF-3DHP. Every number in the table is P-MPJPE, mm. We see that with the increase of the amount of data, the improvement is consistent across all benchmarks.
>
> * **"The paper ... does not include other baselines and previous work for comparison. As the direct related work for comparison is limited, more baselines could be formed... "**
>
> We agree that the comparison against other methods would be beneficial. We discuss existing semi-supervised approaches in the related work. However, as the reviewer stated: "the direct comparison is limited", since in our work, we assume we have access only to the unlabelled videos and extract useful supervision directly from them.
>
> Preserving the defined setting, we suggest to use an off-the-shelf predictor, such as a *2D keypoint* estimator as an alternative baseline. Although such model requires large amount of body-related data during its training, which violates our primary intentions of using less supervision, we can treat it as a black-box model that brings additional signal and apply it in our training pipeline to unannotated video frames. Beyond that, one can also combine the proposed optical flow with the 2D keypoint supervision to show their complementarity.
>
> We use the open-source [Detectron2](https://github.com/facebookresearch/detectron2) model for this purpose. Following the optimization in Eq. 4, we take the loss to be the combination of fully supervised $L_{sup}$ and the additional $L_{2D}$ alignment between 2D keypoints predicted by off-the-shelf 2D pose estimator and the joints extracted from the predicted body shape. As we do with TexturePose, we repeat the experiment by minimizing a loss that combines ours and 2D keypoint alignment (as in Eq. 7).
>
> We repeat the procedure described in Sec. 4.2. The numbers in the table below can be seen as the extension of Table 1, where we add the comparison against 2D keypoint supervision (2D) and the combination of our method with 2D keypoints (2D + OF).
>
> |         | 10%  | 20%  | 50%  | 100% |
> |---------|------|------|------|------|
> | BL      | 65.3 | 62.2 | 59.4 | 58.2 |
> | OF      | 61.3 | 59.3 | 58.0 | 57.1 |
> | 2D      | 58.4 | 57.1 | 56.4 | 56.0 |
> | 2D + OF | 57.6 | 56.5 | 56.0 | 55.8 |
>
> Even though 2D keypoints encode more body-related information, the optical flow serves a useful additional alignment that enhances the model's performance (see 2D + OF line). This observation underscores the utility of optical flow as an additional source of supervision.
>
> * **"The paper is based solely on the SMPL model."**
>
> We totally agree that the SMPL is not an exclusive candidate for a pose and shape model. While we acknowledge that models like [STAR](https://star.is.tue.mpg.de/) or [GHUM](https://openaccess.thecvf.com/content_CVPR_2020/papers/Xu_GHUM__GHUML_Generative_3D_Human_Shape_and_Articulated_Pose_CVPR_2020_paper.pdf) offer greater articulation and diversity than SMPL, our choice of SMPL was motivated by its status of a common open-source benchmark. Unfortunately, there are limited datasets or baselines for the alternative models.
>
> Our primary focus is on exploring data efficiency in training volumetric models, rather than solving human body pose exclusively. In future work, we plan to extend our approach to more challenging domains, such as animals, where there is a wealth of unlabeled videos but limited models and annotated datasets.

---

> > ### Author Response · Authors · 2023-12-19
> > **Authors Response to Reviewer GTP2 (Cont'd)**
> >
> > * **Table 2 should include the additional number of training images ... and the additional training time to have an overview of the cost of the additional performance.**
> >
> > We thank the reviewer for this suggestion. This is a very good idea to illustrate the cost of the improvement. We have added the number of training images and training time to Table 2 and provide the extended version of the table below.
> >
> > |                     | P-MPJPE | Accel.Err  | #samples | Training time, #iterations |
> > |---------------------|---------|------------|----------|----------------------------|
> > | H36M                | 57.1    | 24.4       | 293k     | 150k                       |
> > | H36M + 3DPW-tr      | 56.4    | 24.0       | 316k     | 160k                       |
> > | H36M + 3DPW-tr + PA | 56.2    | 23.9       | 478k     | 270k                       |
> > | 3DPW-test           | 55.8    | 20.2       | 34k      | 30k                        |
> >
> > The "#samples" column denotes the number of additional training samples, and the "#iterations" shows the number of the network weights updates needed for model convergence during fine-tuning.

---

> > > ### Comment · Reviewer_GTP2 · 2024-01-10
> > > **Final Decision**
> > >
> > > The comments to all reviews should be integrated in the manuscript, in particular the new findings of the experiments.

---

> > > > ### Author Response · Authors · 2024-01-16
> > > > **Manuscript updated**
> > > >
> > > > We have integrated comments to all reviews in the manuscript. All changes are marked with red.

---

### Review · Reviewer_C5dk · 2023-11-21

**Summary Of Contributions:**

This work proposes a novel technique for training 3D human pose estimation methods by supplementing labelled 2D data with unlabelled videos of humans. The approach works by first taking consecutive frames of videos, computing two sets of 2D pixel motion for the human (1) with an off-the-shelf optical flow method and (2) by projected mesh vertices of a 3D pose estimator outputting a SMPL model. Given these two sets of estimates, a loss is derived to encourage the visible projected vertices to follow the optical flow estimation.

The quality of using optical flow as a source of supervision is validated through analysis (Sec. 4.6) and empirically by showing improvements in pose estimation error when added to a baseline pose estimation method. The positive impact of the proposed technique is greater when there is less labelled 2D data available.

**Audience:**

Yes

**Broader Impact Concerns:**

There are no broader impact concerns.

**Claims And Evidence:**

Yes

**Requested Changes:**

To improve the justification of the proposed technique, I suggest the authors be precise about the sources of 3D pose estimation training data that are available and are unrestricted because of licensing problems, and potentially demonstrate how their technique practically allows for these restrictions to be overcome.

**Strengths And Weaknesses:**

*Strengths*
- The presentation and writing are exemplary. The method is very well motivated, explained and justified. Code is provided in the supplement so reproducibility is not a concern.
- Using optical flow methods as an additional source of supervision is well motivated, and this work implements this idea effectively. The novel technique can be applied as an additional loss term for any 3D pose estimation method that outputs SMPL parameters.
-  The proposed approach works for single images or videos at test time. This is possible because of a small MLP that is trained to output an affine transform for the current output of the image encoder, given N previous SMPL parameter estimates. Table 3 empirically validates the effectiveness.
- The proposed approach decreases pose estimation error, especially in the low data regime. Adding additional unannotated data further improves performance.

*Weaknesses*
- Despite the motivation to improve 3D human pose estimators using unlabelled video data, the proposed approach is demonstrated to be effective but lags behind the state of the art. It would be a great improvement of the paper if the proposed technique for supervision is shown to positively benefit state of the art methods.
- When 100% of labeled 2D data is available, the relative improvements appear to be marginal. The authors explain that the proposed method is still useful because of licensing and privacy restrictions. It would be beneficial to be explicit here.

---

> ### Author Response · Authors · 2023-12-19
> **Authors Response to Reviewer C5dk**
>
> We sincerely appreciate the valuable comments and feedback provided by the reviewer ${\color{orange} C5dk}$. Also, we would like to thank them for noting the inclusion of our code in the supplement. Please, find our response to each of your concerns below.

---

> > ### Author Response · Authors · 2023-12-19
> > **Authors Response to Reviewer C5dk (Cont'd)**
> >
> > * **"... the proposed approach is demonstrated to be effective but lags behind the state of the art. It would be a great improvement of the paper if the proposed technique for supervision is shown to positively benefit state of the art methods."**
> >
> > Our work investigates the potential of utilizing new data sources as partial substitutes for explicit supervision. In Sec 4.5 and Fig. 5, we demonstrate the performance enhancement for both ResNet50 and HRNet-W32 backbones, even when trained with 100% of data. Both of these backbones essentially constitute strong baselines and former state-of-the-art models of [EFT](https://arxiv.org/pdf/2004.03686.pdf) and [PARE](https://github.com/mkocabas/PARE). In case of the former, combining our semi-supervised approach with only 50% of full-body supervision beats out the model trained with 100% of the data, which reflects the usefulness of the optical flow as additional source of supervision.
> > We also fine-tune the most recent state-of-the-art transformer-based model [HMR2.0](https://shubham-goel.github.io/4dhumans/) on the video sequences we have, however, without noticeable improvement. We attribute this to the fact that some datasets, such as Human3.6M have already been used in this model for training, and the model is already too powerful.
> >
> > We would like to emphasize that our primary focus is not to directly compete with state-of-the-art models but to highlight alternative data sources with potential exploitation. While we acknowledge the prevailing "data is king" paradigm, we do believe our method represents a valuable step toward data-efficient approaches.
> >
> > * **"... The authors explain that the proposed method is still useful because of licensing and privacy restrictions. It would be beneficial to be explicit here. ... I suggest the authors be precise about the sources of 3D pose estimation training data that are available and are unrestricted because of licensing problems, and potentially demonstrate how their technique practically allows for these restrictions to be overcome."**
> >
> > We appreciate the reviewer for pointing out the statement regarding licensing and privacy concerns. We find this crucial to clarify it. In our work, we assert that privacy and licensing concerns drive the pursuit of less data-efficient techniques, as our method is aiming to be. To be precise, let us distinguish between *licensing* and *privacy*, as they have slightly distinct impacts on the research process.
> >
> > *Licensing*: Commonly used research datasets typically carry a 'For Academic Use Only' license. Sometimes, such licenses might expire, hindering the scientific progress, as happened with [Human3.6M Mosh](https://github.com/akanazawa/human_dynamics/blob/master/doc/train.md#note-on-data).
> > All datasets utilized in the present work are under this academic license. This makes such data ineligible for training models intended for commercial use. It drives companies to either spend resources for their own data acquisition or to develop other approaches that require less supervision. Our method exemplifies the latter. It offers a solution to partially alleviate the requirement for explicit body-related supervision that is expensive to acquire and redundant, as demonstrated in our work (Sec. 4.2 and Fig. 3).
> >
> > *Privacy*: Practically, all human-related datasets are compositions of actions, which are acquired via capturing *individual* subjects. It adds potential legal risks for further use. For example, such individuals might not know that their personal data is used (for example, there are multiple "in-the-wild" datasets with videos fetched from Youtube or Instagram, such as [InstaVariety](https://github.com/akanazawa/human_dynamics/blob/master/doc/insta_variety.md)). This brings another valuable motivation to explore various sources of useful data that are less personified. A potential solution is to leverage synthetic data (where such risks are lower), as demonstrated in our work, utilizing optical flow learned without the use of any personal information.
> >
> > It is worth to clarify a section in the introduction where we state: "Unfortunately, privacy and licensing issues still may limit the usage of networks trained in this manner. This motivates our search for less data-intensive techniques." We acknowledge that this sentence may be misleading, as if we claimed that our method allows overcoming privacy and licensing issues for models already trained with licensed data. We propose replacing this section with the following: "Unfortunately, privacy and licensing issues still may limit the usage of these data for training models intended for commercial use, whereas the custom data acquisition can be troublesome. This motivates our search for more data-efficient techniques."

---

### Review · Reviewer_LJLC · 2023-12-13

**Summary Of Contributions:**

This work considers the task of pose and body-shape estimation (i.e. estimating SMPL parameters) from images. It proposes a new training method incorporating self supervision from unannotated video data. Here a standard, pre-trained optical flow model (RAFT) is run on the video frames, and this estimated flow is used to regularise the predicted poses from adjacent frames to be consistent. The proposed method is demonstrated using a CNN backbone with a mixture of labelled and unlabelled data from several sources; the self-supervised training is shown to significantly improve on supervised training on the labelled data only, and also to improve on another self-supervised method.

**Audience:**

Yes

**Broader Impact Concerns:**

None -- this work does not raise any signficant concerns (nor is there any broader impact statement present)

**Claims And Evidence:**

No

**Requested Changes:**

- see the points under weaknesses – in particular, some of the claims need adjusting (or, preferably, evidencing properly)
- give some more convincing motivation (e.g. find some genuinely-low-data scenario where the method is useful and performs well)
- typo – p4 – "visibilty"

**Strengths And Weaknesses:**

Strengths
- the proposed approach is novel. It is clearly and sensibly motivated, and the components are described
- the specific idea of using optical flow to directly regularise the vertex displacements of a SMPL model is straightforward and elegant
- the method works mainly in the setting of image-to-pose, but there is also an extension presented for the video-to-pose setting, requiring only small modifications (and still using the original per-image network without retraining)
- the evaluation is detailed and largely comprehensive, covering the proposed method and one self-supervised baseline (as well as their combination), across different supervision levels
- the evaluation uses different video datasets for the self-supervised part, and shows that increasing amounts of video data from diverse sources improves the model performance
- the writing is generally very good – clear, literate, and pleasant to read; there are very few typos

Weaknesses
- saying (relatively old) TexturePose is only relevant competitor because others rely on keypoints is a little hopeful – in particular one could use such a method with 2D keypoints predicted from a pretrained model (in much the same way that the proposed work uses optical flow predicted from a pretrained model)
- 4.1 says use a resnet because newer, fancier models require lots of training data – but the method is explicitly sold in the introduction as improving on the best ("our approach can even improve the results of state-of-the-art networks")
- also the statement in the introduction that "Our method delivers these benefits no matter what network architecture is used" seems to be contradicted in 4.1. Either should actually show it working with various recent architectures, or should not make this claim
- the part on prediction-from-video is somewhat 'detached' from the rest of the work. There is plenty of literature on this task, yet the proposed approach simply presents a straightforward addition to the per-image model that allows it to work on video. Instead, if prediction from video is a goal, I would have expected authors to use an alternative, spatiotemporal network (e.g. a 3D CNN) for this setting, and to show that when trained with the proposed self-supervised loss, performance is better than without
- all the experiments use datasets that actually have keypoint annotations available; hence the argument that the method is important in the 'practical' setting of limited data is weak. The paper would be much stronger if it showed (or at very least described) a use-case where there is much more limited ground-truth available, and this ground-truth is harder to obtain (in contrast with humans where pretrained models can create respectably good pseudo-labels for in-the-wild datasets).

---

> ### Author Response · Authors · 2023-12-19
> **Authors Response to Reviewer LJLC**
>
> We thank the reviewer ${\color{purple} LJLC}$ for their thorough review and instructive feedback on our paper. Also, we highly appreciate that they find our approach novel and elegant.
> We have carefully addressed all the concerns and suggestions, please, find our response to each of your points below.
>
> ______________________
>
> * **"saying (relatively old) TexturePose is only relevant competitor because others rely on keypoints is a little hopeful – in particular one could use such a method with 2D keypoints predicted from a pretrained model (in much the same way that the proposed work uses optical flow predicted from a pretrained model)"**
>
> The reviewer raises an important issue of providing a more comprehensive comparison. As we focus on extracting useful supervision solely from unannotated videos, it limits our ability to compare with existing semi-supervised approaches discussed in related work. Yet, we consider the reviewer's suggestion to use a 2D pose estimator highly reasonable. Subsequently, we utilize an off-the-shelf 2D keypoint estimator and extract 2D poses from unannotated video frames and minimize the keypoint alignment with these data. This aligns well with our setting, assuming such an extractor is a black-box model providing additional signal (as in Eq. 4 and Eq. 6). Additionally, we combine the proposed optical flow with 2D keypoint supervision to demonstrate their complementarity (as in Eq. 7).
> We take an open-source [Detectron2](https://github.com/facebookresearch/detectron2) to extract 2D pose estimates.
>
> We repeat the procedure described in Sec. 4.2. The numbers in the Table below can be seen as the extension of Table 1, where we add the comparison against 2D keypoint supervision (2D) and the combination of our method with 2D keypoints (2D + OF).
>
> |         | 10%  | 20%  | 50%  | 100% |
> |---------|------|------|------|------|
> | BL      | 65.3 | 62.2 | 59.4 | 58.2 |
> | OF      | 61.3 | 59.3 | 58.0 | 57.1 |
> | 2D      | 58.4 | 57.1 | 56.4 | 56.0 |
> | 2D + OF | 57.6 | 56.5 | 56.0 | 55.8 |
>
> It is important to highlight that a 2D keypoint estimator of this kind requires a significant amount of body-related supervision during training, contradicting our goal of using less supervision. No wonder that it brings a larger improvement (2D line) compared to the synthetic optical flow model we employ (OF line). Nevertheless, combining 2D keypoints with optical flow enhances the model's performance (2D + OF line). This observation underscores the utility of optical flow as an additional source of supervision.
>
>
> * **"4.1 says use a resnet because newer, fancier models require lots of training data – but the method is explicitly sold in the introduction as improving on the best ("our approach can even improve the results of state-of-the-art networks")"**
>
> * **"also the statement in the introduction that "Our method delivers these benefits no matter what network architecture is used" seems to be contradicted in 4.1. Either should actually show it working with various recent architectures, or should not make this claim"**
>
> In Sec. 4.5 and Fig. 5, we demonstrate the improvement over ResNet50 and HRNet-W32 backbones, even when trained with 100% of data. Both of these backbones essentially constitute strong baselines and former state-of-the-art models of [EFT](https://arxiv.org/pdf/2004.03686.pdf) and [PARE](https://github.com/mkocabas/PARE).
> We also make additional tests, where we fine-tune the most recent state-of-the-art transformer-based model [HMR2.0](https://shubham-goel.github.io/4dhumans/), however, without significant improvement. We attribute this to the fact that some datasets, such as Human3.6M have already been used in this model for training.
>
> Nevertheless, our approach is able to boost the performance of baseline models of various backbones. Hence, we believe that our phrase in the introduction "our approach can even improve the results of state-of-the-art networks" is not far from reality, yet we propose to replace it with the one that is more correct: "our approach can improve the results of strong baseline models of various architectures".
>
> It should be noted that the sentence in Sec 4.1 "Hence, they [more sophisitcated architectures] are not suitable for our main experiments." is not entirely correct. In Sec. 4.5 we demonstrate that our method can be applied to HRNet-W32, though our "main experiments" are low-data regime. We propose to change the phrase by saying "Hence, they are less suitable for our main experiments in the low-data regime, yet we demonstrate that our method can be applied to more sophisticated architectures, such as HRNet-W32."
>
> At the same time, we totally agree with the reviewer that some phrases in the introduction are too strong. We propose to replace "Our method delivers these benefits no matter what network architecture is used" with "Our method delivers these benefits for various network architectures", which is true, as shown in Sec. 4.5.

---

> > ### Author Response · Authors · 2023-12-19
> > **Authors Response to Reviewer LJLC (Cont'd)**
> >
> > * **"the part on prediction-from-video is somewhat 'detached' from the rest of the work. There is plenty of literature on this task, yet the proposed approach simply presents a straightforward addition to the per-image model that allows it to work on video. Instead, if prediction from video is a goal, I would have expected authors to use an alternative, spatiotemporal network (e.g. a 3D CNN) for this setting, and to show that when trained with the proposed self-supervised loss, performance is better than without"**
> >
> > We understand the reviewer's concern and would like to emphasize that prediction-from-video is not our focus. We do not try to obtain a better video-based model and compete with state-of-the-art methods that utilize specific temporal architectures. Instead, we simply demonstrate how to exploit temporal information that is naturally encoded in the optical flow, and in Sec. 4.4 we show how it makes the single-frame model compatible with the video inputs.
> >
> > In our experiments, we tried to learn temporal layers, such as GRU or LSTM, using only optical flow as the supervision for videos. As we mentioned in the paper, this works poorly not just on single-frame pose and shape estimation, but also on video inputs - optical flow is too weak to efficiently learn temporal architectures.
> >
> > * **"all the experiments use datasets that actually have keypoint annotations available; hence the argument that the method is important in the 'practical' setting of limited data is weak. The paper would be much stronger if it showed (or at very least described) a use-case where there is much more limited ground-truth available, and this ground-truth is harder to obtain (in contrast with humans where pretrained models can create respectably good pseudo-labels for in-the-wild datasets)."**
> >
> > We absolutely agree with the reviewer that the domain of the human body pose estimation, where we have plenty of data, might not suit well for the practical evaluation of our method. We would like to highlight that our primary focus is on exploring data efficiency, rather than solving human body pose exclusively, and we use human datasets exactly because we have a lot of datasets available for evaluation.
> >
> > In principle, our optical flow-based technique could be applied for other moving objects. In future work, we plan to extend our approach to a very challenging domain of animal pose and shape estimation. There is an abundant amount of unannotated videos, yet only a few methods and models that tackle this subject, exactly because there is very little annotated data for this task.
> >
> > * **"typo – p4 – "visibilty""**
> >
> > We thank the reviewer for pointing out this typo and we fix it.

---

### Author Response · Authors · 2023-12-19
**Response with concise summary**

We thank all the reviewers for their invaluable feedback. We are glad that the reviewers find our method novel and elegant (${\color{purple} LJLC}$), components and presentation well written and pleasant to read, easy to follow, clearly explained, motivated and justified (${\color{green} GTP2}$, ${\color{orange} C5dk}$, ${\color{purple} LJLC}$). They note that the related work is complete (${\color{green} GTP2}$), the evaluations are detailed and largely comprehensive (${\color{purple} LJLC}$), and the results are promising and effective (${\color{green} GTP2}$, ${\color{orange} C5dk}$). Also, they regard the code we provided for reproducibility purposes (${\color{orange} C5dk}$).

_____________________________________________________________

Here we present a concise summary of our responses to the reviewers' comments:
- Add more evaluations, on Human3.6M and MPI-INF-3DHP (${\color{green} GTP2}$)
- Discuss our competitors and form another baseline to compare (${\color{green} GTP2}$, ${\color{purple} LJLC}$)
- Discuss our method in regard to the state-of-the-art models (${\color{orange} C5dk}$, ${\color{purple} LJLC}$)
- Discuss our method in regard to the licensing and privacy restrictions (${\color{orange} C5dk}$)
- Clarify the motivation behind the temporal extension (${\color{purple} LJLC}$)
- Express our motivation more concisely (${\color{green} GTP2}$, ${\color{purple} LJLC}$)
- Extend Table 2 with the number of additional images and training time (${\color{green} GTP2}$)

The details on each comment are provided in our responses for each of the reviewers.

---

### Decision · Action_Editor_BkWk · 2024-01-19

**Recommendation:** Accept as is

**Comment:**

Overall all reviewers found the paper to be of sufficient interest to the TMLR community and a useful addition to the literature so I am happy to recommend acceptance.

**Audience:**

3d perception is very much a hot topic in vision which forms a sufficiently large part of the TMLR audience. The paper proposes a way to alleviate dependency on expensive 3d ground truth data, which may appeal to researchers interested in semi-supervised/self-supervised learning.

**Claims And Evidence:**

Claims and evidence are mostly adequate -- only one reviewer asked for the claims in the introduction to be weakened.
After rebuttal, all reviewers are happy with experimental results.

---

> ### Author Response · Authors · 2024-02-06
> **Thank you to AE and Reviewers**
>
> We deeply appreciate the constructive feedback and insightful guidance provided by the action editor and reviewers. Thanks for helping strengthen our work and contributions.
>
> We just posted the camera-ready version of the paper.